# Post-Mortem Analysis of Neuropathological Changes in Human Tinnitus

**DOI:** 10.3390/brainsci12081024

**Published:** 2022-08-01

**Authors:** Faris Almasabi, Faisal Alosaimi, Minerva Corrales-Terrón, Anouk Wolters, Dario Strikwerda, Jasper V. Smit, Yasin Temel, Marcus L. F. Janssen, Ali Jahanshahi

**Affiliations:** 1Department of Neurosurgery, Maastricht University Medical Center, 6229 HX Maastricht, The Netherlands; f.almasabi@maastrichtuniversity.nl (F.A.); f.alosaimi@maastrichtuniversity.nl (F.A.); y.temel@maastrichtuniversity.nl (Y.T.); 2School for Mental Health and Neuroscience, Faculty of Health, Medicine and Life Sciences, Maastricht University, 6229 ER Maastricht, The Netherlands; m.corralesterron@student.maastrichtuniversity.nl (M.C.-T.); aat.wolters@student.maastrichtuniversity.nl (A.W.); d.strikwerda@student.maastrichtuniversity.nl (D.S.); jasper.smit@maastrichtuniversity.nl (J.V.S.); m.janssen@maastrichtuniversity.nl (M.L.F.J.); 3Department of Physiology, Faculty of Medicine, King Khalid University, Abha 62529, Saudi Arabia; 4Department of Physiology, Faculty of Medicine, King Abdulaziz University, Rabigh 25732, Saudi Arabia; 5Department of Ear, Nose, Throat, Head and Neck Surgery, Zuyderland Medical Center, 6419 PC Heerlen, The Netherlands; 6Department of Clinical Neurophysiology, Maastricht University Medical Center, 6229 HX Maastricht, The Netherlands

**Keywords:** tinnitus, serotonin, post-mortem histology, auditory pathway

## Abstract

Tinnitus is the phantom perception of a sound, often accompanied by increased anxiety and depressive symptoms. Degenerative or inflammatory processes, as well as changes in monoaminergic systems, have been suggested as potential underlying mechanisms. Herein, we conducted the first post-mortem histopathological assessment to reveal detailed structural changes in tinnitus patients’ auditory and non-auditory brain regions. Tissue blocks containing the medial geniculate body (MGB), thalamic reticular nucleus (TRN), central part of the inferior colliculus (CIC), and dorsal and obscurus raphe nuclei (DRN and ROb) were obtained from tinnitus patients and matched controls. Cell density and size were assessed in Nissl-stained sections. Astrocytes and microglia were assessed using immunohistochemistry. The DRN was stained using antibodies raised against phenylalanine hydroxylase-8 (PH8) and tyrosine-hydroxylase (TH) to visualize serotonergic and dopaminergic cells, respectively. Cell density in the MGB and CIC of tinnitus patients was reduced, accompanied by a reduction in the number of astrocytes in the CIC only. Quantification of cell surface size did not reveal any significant difference in any of the investigated brain regions between groups. The number of PH8-positive cells was reduced in the DRN and ROb of tinnitus patients compared to controls, while the number of TH-positive cells remained unchanged in the DRN. These findings suggest that both neurodegenerative and inflammatory processes in the MGB and CIC underlie the neuropathology of tinnitus. Moreover, the reduced number of serotonergic cell bodies in tinnitus cases points toward a potential role of the raphe serotonergic system in tinnitus.

## 1. Introduction

Tinnitus is a sensory disorder in which a phantom sound is perceived by affected individuals. The prevalence of tinnitus in Europe is approximated at 15%, while bothersome tinnitus has a prevalence of 6% and severe tinnitus 1–2%, which is less than one out of ten persons with any form of tinnitus. [1]. In 1.2% of the population, tinnitus complaints severely affect the attention, mood, and sleep of patients [1]. Individuals with bothersome tinnitus often have comorbid psychological complaints such as anxiety or depressive symptoms [2]. Although tinnitus has a considerable impact on the individuals as well as society, the exact pathophysiology of tinnitus remains largely unknown. The starting point of tinnitus is thought to be hearing loss that causes peripheral deafferentation [3,4]. Central maladaptive plasticity may occur in response to depleted peripheral input and create a hyperactive state within the central auditory network [5]. A major objective of tinnitus research is to define the neural changes in the central nervous system that underlie the disease. Findings in animal models of tinnitus are crucial in this endeavor, as neural changes can be correlated with behavioral evidence of tinnitus. It is important, however, that findings in animal studies are confirmed in humans to show the validity of such outcomes [6].

Several pathological processes at cellular levels have been reported in animal models of tinnitus, such as an increased spontaneous firing rate in several auditory regions [7]. Upregulation of the neuronal activity marker (c-Fos) has also been reported in auditory and non-auditory regions [8,9]. This hyperactive state may be due to either reduced inhibitory or increased excitatory neurotransmission or changes in neuronal excitability. In the auditory network, reduction in GAD expression, a GABAergic marker, has been reported in tinnitus in the ventral cochlear nucleus (VCN), dorsal cochlear nucleus (DCN), and auditory cortex (AC) [10,11,12,13]. Increased vesicular glutamate transporter expression, a glutamatergic marker, was also evident in the DCN [13,14]. In an ex vivo spectroscopy study in a chronic noise-induced rat model of tinnitus, ambient glutamate was found to be increased in the DCN but decreased in the medial geniculate body (MGB) [15]. Additionally, noise trauma was shown to significantly reduce the cell density in all subdivisions of the MGB and AC [16]. Another study in mice also described a reduction in cell density throughout the auditory network, including the DCN, VCN, the central nucleus of the inferior colliculus (CIC), MGB, and AC, one week after exposure to noise trauma [17]. A number of human imaging studies report anatomical changes in various brain regions related to tinnitus and hearing loss. However, the outcomes are inconclusive [15,16,17]. Substantial effects in the AC [18,19] or subcortical auditory structures such as the inferior colliculus (IC) and MGB have been described [20,21,22]. Specifically, grey matter volumes have been reported to be increased in the MGB and decreased in the IC [20,21,22], but later neuroimaging studies have not been able to replicate these findings [23,24,25,26]. Changes in white matter integrity have also been described [15]. Nevertheless, to date, there is no consensus on anatomical changes in grey matter and/or white matter tracts in the auditory brain network related to the tinnitus pathophysiology [27]. A potential confounder that might explain the opposing results is the co-occurrence of hearing loss.

Non-auditory regions are also thought to play a role in tinnitus neuropathology. Serotonergic impairment is among the early theories of tinnitus pathophysiology [28], as it is known to play a key role in stress, depression, anxiety, and sleep disorders [29]. Interestingly, these are common comorbidities in tinnitus patients [30]. A clinical study has shown that serotonin agonists such as selective serotonin reuptake inhibitors (SSRIs) can reduce tinnitus severity and perceived loudness [31]. Although the evidence suggests shared pathological processes, the serotonergic system has been granted little to no attention when studying tinnitus pathophysiology. The brain stem dorsal raphe nucleus (DRN) is the major source of serotonin in the mammalian brain and sends serotonergic projections to all auditory regions, among other areas [32,33,34]. Experimental evidence suggests that the DRN is affected in tinnitus. For example, serotonergic neuronal activity has been shown to be enhanced in the DRN in a sodium salicylate animal model of tinnitus [35,36], while a microdialysis study reported increased extracellular serotonin levels in the IC and AC after salicylate application [37].

In this post-mortem study, we provide the first evidence on histopathological changes in brain specimens obtained from tinnitus patients and control subjects. Notably, no objective information was available regarding hearing loss in both the tinnitus and control groups. This should be taken into consideration when interpreting the presented data. We measured cell number and size in thalamic regions, including the MGB, as well as the CIC in Nissl-stained sections. We further utilized immunohistochemical markers to examine the astroglia cell population, serotonin, and dopamine-containing cells in the DRN and raphe obscurus nucleus (ROb).

## 2. Materials and Methods

### 2.1. Specimens

Formalin-fixed tissue blocks were obtained from two brain banks: the Netherlands Brain Bank and the Parkinson’s United Kingdom [38] Brain Bank. Diagnosis of tinnitus was made based on brain banks’ selection criteria and a retrospective review of medical records. Only subjects who sought health aid for their tinnitus were included, with exclusion of cases with Ménière’s disease or pulsatile tinnitus. In addition, subjects with neurodegenerative disorders were excluded. No information, such as audiogram, on the hearing performance of both the tinnitus and control group was available. The two populations could thus not be matched according to the degree of hearing loss. The causes of death included generally age-related causes such as heart attack, cancer, and pneumonia. In some control cases, these data were not available. Ethical approval was acquired from the medical ethics committee of Maastricht University Medical Center (METC number 2020-1550).

### 2.2. Histology and Immunohistochemistry

Paraffin-embedded blocks were cut into seven-µm-thick sections on a microtome (LKB Instruments, Bromma, Sweden) and mounted on microscope slides. A serial of ten sections per subject—with 140 µm inter-slice distance—were used for histological and immunohistochemical evaluation, except for glial cell evaluation, where three sections per subject were used with an inter-slice distance of 560 µm. It is important to note that we subjected adjacent sections of every specimen to different histologic assessments to rule out potential bias that could be caused by variation in tissue properties or anatomical levels.

#### 2.2.1. Cresyl Violet (Nissl) Staining

To assess neuronal cell density and size, standard Nissl staining was performed. First, all sections were deparaffinized in Xylene and rehydrated in graded ethanol and phosphate-buffered saline (PBS, 3 × 20 min). Thereafter, sections were immersed twice in glacial acetic acid solution and one time in triton/ethanol solution for 20 min. This was followed by incubation in 0.1 g Cresyl Violet solution for 35 min at 50 °C, and then immersion for one minute three times in glacial acetic acid solution and one time in 100% ethanol. Lastly, sections were dehydrated in Xylene and coverslipped with Entellen.

#### 2.2.2. Immunohistochemistry

Following deparaffinization and rehydration, a microwave antigen retrieval was conducted in boiling citric acid buffer (10 mmol, pH 6.0) for 10 min and then cooled down at room temperature (RT) for 20 min. Sections were washed in Tris-buffered saline (TBS) and blocked for endogenous peroxidase activity with 0.3% H_2_O_2_ in TBS for 20 min. After washing in TBS, sections were incubated for 30 min in TBS-triton (TBS-T) with 3% normal donkey serum (NDS). Primary antibodies were diluted in TBS-T 0.3% NDS and incubated as follows: anti-glial fibrillary acidic protein (GFAP, Agilent Dako, Galstrup, Denmark, Z0334, 1:2000), ionized calcium-binding adaptor protein-1 (iba-1, FUJIFILM Wako, Osaka, Japan, 019-19741, 1:500), anti-phenylalanine hydroxylase subtype 8 (PH8, Santa Cruz Biotechnology, Santa Cruz, CA, United States, sc-58398, 1:500), anti-tyrosine hydroxylase (TH, abcam, Darmstadt, Germany, ab1542, 1:500) for one, two, seven, and seven nights, at 4 °C, 4 °C, RT, and RT, respectively. Afterward, sections were washed and incubated with secondary antibodies (biotinylated donkey (anti-rabbit, anti-rabbit, anti-mouse, anti-sheep, respectively), Jackson laboratory, 1:200) for 60 min at RT. Sections were then washed and incubated for 90 min in avidin–biotin peroxidase complex (Elite ABC kit, Vectastain, Burlingame, CA, USA, 1:50), followed by 10 min incubation in 3,3-diaminobenzidine (DAB, 5 mL DAB, 5 mL Tris–HCl, 3.35 µL H_2_O_2_, with added Nickel chloride (NiCl2) intensification for TH staining only). The DAB reactions were stopped by rinsing all sections in milli-Q (2 × 3 min), followed by dehydration, and samples were finally coverslipped with Entellen.

### 2.3. Quantification

Researchers were blinded to experimental groups. Due to low thickness of the tissue, stereological cell counting could not be achieved. Instead, a systematic semi-quantification analysis was performed. To avoid potential bias from cutting angles and variation in the size of investigated regions across subjects, quantification of cells was divided by the counted area to obtain cell density in Nissl-, PH8-, and TH-stained specimens.

#### 2.3.1. Defined Regions of Interest

The regions of interest (ROI) included the MGB, thalamic reticular nucleus (TRN), CIC, DRN, and the ROb. The lateral nucleus of pulvinar (LP), which is part of the visual circuitry, was included as a control region [39]. The ROIs were delineated based on the Allen Human Brain Atlas [40]. For MGB, two anatomical levels were selected: the rostral level or limitans part of the MGB, which is a semi-oval shape in close proximity to the substantial nigra (Figure 1A,B), and the caudal MGB, where it becomes a round eminence separated from the LP and lateral geniculate body by white matter (Figure 1C,D). At this level, all MGB subdivisions including the dorsal (MGBd), ventral (MGBv), and medial (MGBm) can be distinguished, whereas the limitans MGB only encompasses the MGBd and MGBv. The TRN is a thin layer of GABAergic cells surrounding the rest of the thalamic nuclei. We performed GAD (GAD 65 + 67, abcam, ab18399, 1:500) staining to define its boundaries from the rest of the thalamic nuclei. The CIC (Figure 1E,F), DRN, and ROb regions were delineated according to the Allen Human Brain Atlas.

#### 2.3.2. Quantification of the Nissl-Stained Cells

Cell density was quantified in the MGBv, MGBd, MGBm, TRN, LP, CIC, and DRN. The ROIs were delineated at 2× magnification and cell counting was performed at 40×. The optical fractionator probe in the stereo investigator program (ver. 2020 1.3, MBF Bioscience, Williston, VT, USA) was used to randomly select 10% of the total counting frames in every delineated ROI. Cells were counted and the average of ten sections divided by delineated area was used for statistical analysis. Cell size was measured in the MGBv, MGBd, MGBm, and CIC using the surfactor probe in the stereo investigator software, based on a previously published method [41]. In brief, by selecting the nucleus in each cell, four emanating lines were automatically drawn by the program and selections were made at the intersection with the borders of each cell. Given its laborious nature, cell size was measured in 2% of the ROIs. We first started with 10% in two randomly selected specimens per group and reduced it to 2% when no difference was observed. The mean value of ten sections was used for statistical analysis.

#### 2.3.3. Quantification of the Glial Cells

Consecutive sections adjacent to Nissl-stained sections were stained with anti-GFAP and iba-1 antibodies to assess the number of astrocytes and microglia cells, respectively. Three sections per subject, at 3 rostrocaudal levels with 560 µm distances, were photographed in an AX-70 microscope (Cell-P software, Olympus Soft Imaging Solutions, Münster, Germany) connected to a motorized condenser (Olympus, Zoeterwoude, The Netherlands). In every CIC- and MGB-containing section, four non-overlapping fields were photographed using a 10× objective. In the DRN, two fields were photographed. In the CIC, images were taken dorsoventrally from the middle of the structure. In the MGB, one image was taken from the MGBm, one from the MGBv, and two from the MGBd. At the limitants MGB level, two images were taken from the MGBd and MGBv. In the DRN, two dorsoventral images were taken ventral to the cerebral aqueduct. Glial cells were counted manually using ImageJ (ImageJ software version 1.52p; National Institutes of Health, Bethesda, MD, USA) after selecting an appropriate grey value threshold for all regions. Only cells that had a characteristic shape (star-shaped and ramified astrocytes and microglia cells, respectively) and were above the grey value threshold were counted. The mean of the three levels was used for statistical analysis.

#### 2.3.4. Quantification of the Serotonergic and Dopaminergic Cells

We used anti-PH8 to stain serotonergic cells, a widely used antibody in human post-mortem studies, as direct staining of the serotonin molecule is nearly impossible in post-mortem tissue [42,43,44]. We also used anti-TH antibody to stain dopaminergic cells in the DRN. Adjacent sections of the DRN were stained with anti-PH8 and TH antibodies to detect serotonin- and dopamine-containing cells, respectively. Additionally, PH8 staining was performed for the medullary raphe, i.e., Rb, which lies at the same level as the first auditory brain area, the DCN. Four tinnitus samples per group were used for the DRN and ROb. Similar to the quantification of Nissl-stained sections, ROIs were delineated at 2× magnification and cells were counted at 40× objective with the optical fractionator probe in the stereo investigator program. The entire stained area was subjected to cell counting. In addition, the morphology of PH8-positive cells was qualitatively assessed.

### 2.4. Statistics

To check for normality, a Shapiro–Wilk test was performed. As a normal distribution was given for all analyzed parameters, an independent-samples t-test was used to detect changes between groups. Data were analyzed using SPSS software (SPSS Version 25.0, IBM, Chicago, IL, USA). *p* values < 0.05 were considered to be statistically significant and all statistical tests were two-tailed. All data are presented as the mean ± standard error of means (SEM). For confounders, we checked if there was a difference between groups in post-mortem delay, sex, or age for all parameters.

## 3. Results

Brain samples from nine tinnitus and eight matched control subjects were included in this study. Table 1 summarizes the subjects’ tinnitus criteria, matched controls, and ROIs. Neither post-mortem delay, age, nor sex differed significantly between groups. Notably, not all regions were available in each subject (Table 1).

### 3.1. Cell Count and Size

We observed a significant reduction in cell density in the MGBd (control (C) = 26.9 ± 2.2, tinnitus (T) = 18.8 ± 2.2, *p* = 0.02), MGBv (C = 25.8 ± 2.4, T = 18.5 ± 2.0, *p* = 0.04), whole MGB (C = 25.9 ± 2.1, T = 18.1 ± 2.0, *p =* 0.02), and CIC (C = 18.1 ± 1.0, T = 13.6 ± 1.3, *p =* 0.02) of tinnitus subjects compared to the controls. No significant changes were found in the TRN (C = 11.2 ± 1.3, T = 9.6 ± 1.4, *p =* 0.43), LP (C = 14.9 ± 2.0, T = 14.5 ± 0.6, *p =* 0.82), or DRN (C = 21.9 ± 1.5, T = 18.5 ± 1.8, *p =* 0.20) between groups. The analysis of cell size did not show any significant change in any investigated region between groups (Figure 2).

### 3.2. Glial Cells

We found a significant reduction in the number of astrocytes in the CIC (C = 131.7 ± 16.5, T = 75.4 ± 11.4, *p =* 0.02) of the tinnitus group when it was compared to the controls. Statistical analysis did not show any significant changes between groups in the MGBd (C = 63.1 ± 8.5, T = 49.5 ± 9.2, *p =* 0.42), MGBv (C = 32.7 ± 4.1, T = 24.5 ± 5.0, *p =* 0.25), MGBm (C = 31.9 ± 4.3, T = 30.0 ± 5.7, *p =* 0.80), whole MGB (C = 127.6 ± 16.3, T = 107.2 ± 18.6, *p =* 0.45), or DRN (C = 76.3 ± 4.2, T = 68.7 ± 12.7, *p =* 0.91) (Figure 3).

Quantification of microglia cells showed no significant difference between groups in any investigated region (CIC (C = 191.9 ± 6.6, T = 186.9 ± 20.3, *p =* 0.82), MGBd (C = 91.2 ± 3.6, T = 89.6 ± 8.1, *p =* 0.54), MGBv (C = 46.0 ± 1.7, T = 52.3 ± 3.3, *p =* 0.15), MGBm (C = 49.9 ± 2.3, T = 50.5 ± 3.8, *p =* 0.90), whole MGB (C = 187.1 ± 6.1, T = 198.5 ± 12.3, *p =* 0.47), or DRN (C = 101.0 ± 5.0, T = 115.8 ± 7.2, *p =* 0.14)) (Figure 4).

### 3.3. Serotonergic Cells

There were significantly fewer PH8-containing cells in the DRN and ROb in the tinnitus group compared to controls (C = 95.5 ± 12.5, T = 13.5 ± 4.2, *p* < 0.001; C = 33.7 ± 3.1, T = 13.6 ± 0.5, *p* < 0.001, respectively; Figure 5 and Figure 6). Moreover, there was a noticeable difference in the morphology between groups. Tinnitus samples exhibited weaker expression of PH8-containing cells with less dense synaptic fibers compared to controls. Demonstrative low-power photomicrographs per subject are included in Figure 5.

### 3.4. Dopaminergic Cells

TH staining was conducted in serial sections adjacent to those used for PH8 staining. Quantification of TH-containing cells showed no difference between groups (C = 16.7 ± 0.4, T = 15.6 ± 1.1, *p =* 0.39) (Figure 5 and Figure 6).

## 4. Discussion

This study provides the first histological evidence on changes in areas relevant to tinnitus in human tissue. We observed reduced cell density in both dorsal and ventral subdivisions of the MGB and the CIC in individuals who had tinnitus in their clinical history. The CIC also contained a lower number of astrocytes in the tinnitus cases. Intriguingly, the numbers of serotoninergic cells in the DRN and ROb were substantially reduced in tinnitus patients compared to the controls, while no change was observed in the number of dopaminergic cells in the DRN. Our data point toward signs of neurodegeneration in tinnitus, indicated by reduced neuronal cell density in auditory regions. Note that the LP (visual thalamus) did not show any reduction in cell density. No significant changes in cell counts were observed in the TRN and DRN. These findings are in line with experimental data obtained from a noise trauma-induced animal model of tinnitus. Two studies reported reduced cell density in the MGB of mice exposed to noise trauma [45,46]. In the same study, Gröschel et al., (2010) also described a reduced cell density in the CIC [46]. Interestingly, recent research suggests an association between tinnitus and neurodegenerative disorders. In a retrospective cohort study with a 10-year follow-up, patients that suffered from tinnitus were 1.5 times more likely to develop Alzheimer’s or Parkinson’s disease compared to control subjects [47]. Therefore, it can be speculated that neurodegenerative processes may play a role in tinnitus pathophysiology. On the other hand, tinnitus has also been suggested to be an early symptom of neurodegenerative disorders [48].

Given the fact that any form of neurodegeneration can affect the non-neuronal cell population, namely the astroglia [49], we sought to address whether these changes are detectable in the auditory pathway in tinnitus patients. No alterations in microglia and astrocyte expression were found in the MGB. In the CIC, however, a reduction in the number of GFAP-expressing cells was found (Figure 3). This finding was surprising, as the common assumption would be that cell death and possibly apoptosis would increase the number of astroglia in the affected areas [50]. On the other hand, the reduction in astrocytes is similar to what is commonly found in mood disorders [51]. Astrocytes have been shown to be consistently reduced in psychological disorders such as chronic stress, anxiety, and depression [52,53,54,55]. Whether these changes are the cause or consequence of neurodegeneration remains to be elucidated. In animal models of tinnitus, opposing findings have been described. In a salicylate model of tinnitus, increased GFAP and Iba-1 expression was reported in the primary AC and in the MGB [56]. In a noise trauma animal model of tinnitus, microglia activity was enhanced in the AC five days after noise trauma. Following pharmacological depletion of the microglia, noise trauma failed to cause tinnitus, suggesting an essential role of microglia in tinnitus pathogenesis [57]. Nevertheless, discrepancies in the inflammatory system’s changes reported in animal studies cannot directly be translated to humans given the acute nature of tinnitus induction in animal models [58].

Finally, we investigated the changes in serotonergic and dopaminergic cell bodies in the brainstem. Serotonergic input to the investigated areas originates mainly from the DRN but also Rob, which projects to the DCN [33]. We found a significant reduction in the number of PH8-containing cell bodies in both regions in the tinnitus group compared to controls. Moreover, PH8-containing cells showed an abnormal morphology in all tinnitus cases (Figure 5). Qualitative inspection revealed that the PH8-containing cells appeared pale (possibly containing less enzyme) and exhibited thinner and shorter axons and dendrites with fewer neuronal arborizations in tinnitus patients. In order to rule out the effect of tissue quality on the immunohistochemical signal, we stained sections adjacent to those that were stained for PH8 with an antibody raised against TH, as the DRN hosts a substantial number of dopaminergic cells. Statistical analysis showed no difference in the number of the DRN TH cells between groups. Moreover, there was no difference in TH cells’ morphology between groups. Therefore, we concluded that the affected serotonergic cell population in the DRN of tinnitus patients was less likely to be related to tissue processing issues such as fixation or post-mortem delay. The involvement of serotonergic systems in chronic tinnitus is poorly studied. Our data, however, are in line with clinical observations in tinnitus patients. The comorbidity scores of anxiety disorders and depression are quite high in tinnitus patients (approximately 45% and 33%, respectively) [59,60]. Sleep disorders are another frequent complaint among severe tinnitus patients [61]. Degeneration of raphe serotonergic neurons is in line with these comorbidities. Moreover, treatment with SSRIs has been shown to improve the quality of life in some tinnitus patients and might influence the perception of tinnitus loudness [31], but such conclusions lack extensive clinical validation. Clinical trials that have been conducted in tinnitus patients carry methodological limitations [62,63]. A downregulated serotoninergic system is in line with the gating dysfunction theory of tinnitus. Based on “noise cancellation” hypothesis, when the auditory regions are at a hyperactive state due to deafferentiation, the “tinnitus signal” should be gated at the thalamic level through inputs from collateral pathways. Based on this model, inputs from limbic and raphe nuclei should trigger the TRN and subsequently inhibit the MGB in order to suppress tinnitus [64]. Moreover, serotonergic inputs could also modulate auditory pathways directly as they impose a predominantly inhibitory effect on regions such as the AC, DCN, and amygdala [65,66,67] that could suppress neuronal hyperactivity in these areas. Therefore, this compensatory mechanism could be impaired when the serotonergic system (DRN) malfunctions.

It remains challenging to disentangle the tinnitus pathophysiology from alterations caused by hearing loss, as most tinnitus patients have some degree of hearing loss. In particular, in this study, it was not possible to determine whether any hearing loss contributed to tinnitus. This post-mortem study consisted of a relatively small sample size when considering tinnitus heterogeneity among individuals. Stereological quantification of stained tissue could compensate for the small sample size, but due to the low thickness of paraffin tissue sections, we were not able to apply this method. As a subjective “sensory” disorder, tinnitus diagnosis is made based on patient self-reporting. For those who seek health services for their tinnitus, which was the defining factor of our sample, tinnitus scores are expected to be higher than non-seekers [68]. Nevertheless, tinnitus severity could not be defined further in our subjects, as it would require full access to tinnitus and comorbidity scores. Comorbidities such as depression and anxiety could have additionally contributed to pathological changes in this study.

## 5. Conclusions

We observed a reduction in cell density in key auditory areas, namely the MGB and CIC, in post-mortem tinnitus tissue. The CIC also contained a lower number of astrocytes, whereas the microglia population was unaffected. These findings suggest that neurodegenerative processes may play a key role in tinnitus neuropathology. Moreover, the reduced number of serotonergic neurons in tinnitus cases points towards a role of the raphe serotonergic system in this complex neurological disorder.

## Figures and Tables

**Figure 1 brainsci-12-01024-f001:**
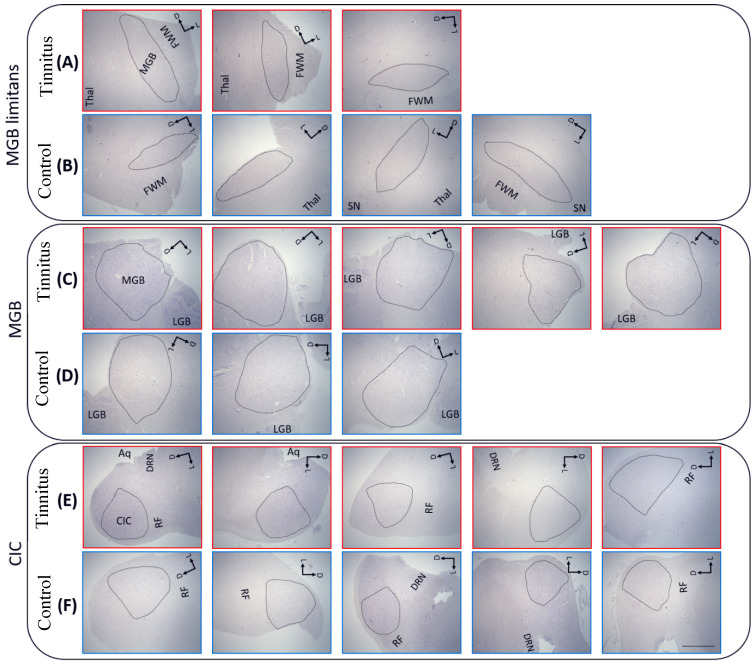
Representative low-power photomicrographs of brain sections per subject stained with Nissl, showing the medial geniculate body (MGB; (**A**–**D**)) and central part of inferior colliculus (CIC; (**E**,**F**)). Two different anatomical levels of the MGB were included: the limitans (**A**,**B**) and the more caudal round region (**C**,**D**). Tinnitus subjects are shown in (**A**,**C**,**E**) and controls in (**B**,**D**,**F**). Scale bar = 2 mm; Aq, cerebral aqueduct; D, dorsal; DRN, dorsal raphe nucleus; FWM, frontal white matter, L, lateral; LGB, latera geniculate body; RF, reticular formation; SN, substantial nigra; and Thal, thalamic nuclei.

**Figure 2 brainsci-12-01024-f002:**
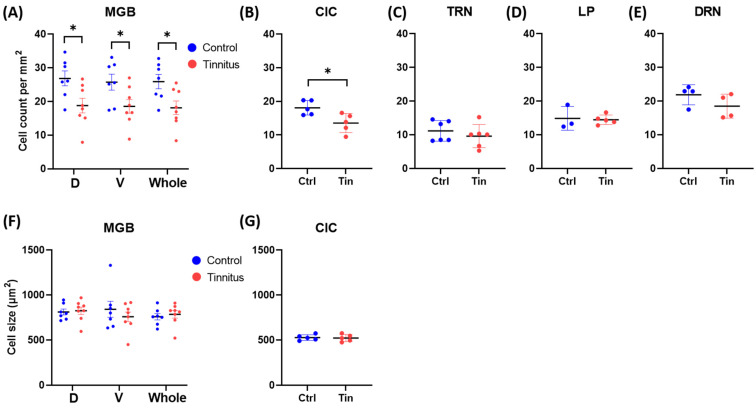
Cumulative data showing the mean cell density (**A**–**E**) and cell size (**F**,**G**) measured in Nissl-stained sections containing the anatomical areas. (**A**,**F**) graphs show the cell density and size, respectively, in (D) dorsal, (V) ventral, and MGB (as a whole). *p* value < 0.05 was defined as significant level and is indicated by an (*). Note the significant reduction in cell density in the MGB and its subdivisions (**A**), as well as in the CIC (**B**). CIC, central part of inferior colliculus; Ctrl, control group; DRN, dorsal raphe nucleus; LP, lateral nucleus of pulvinar; MGB, medial geniculate body; Tin, tinnitus group; and TRN, thalamic reticular nucleus.

**Figure 3 brainsci-12-01024-f003:**
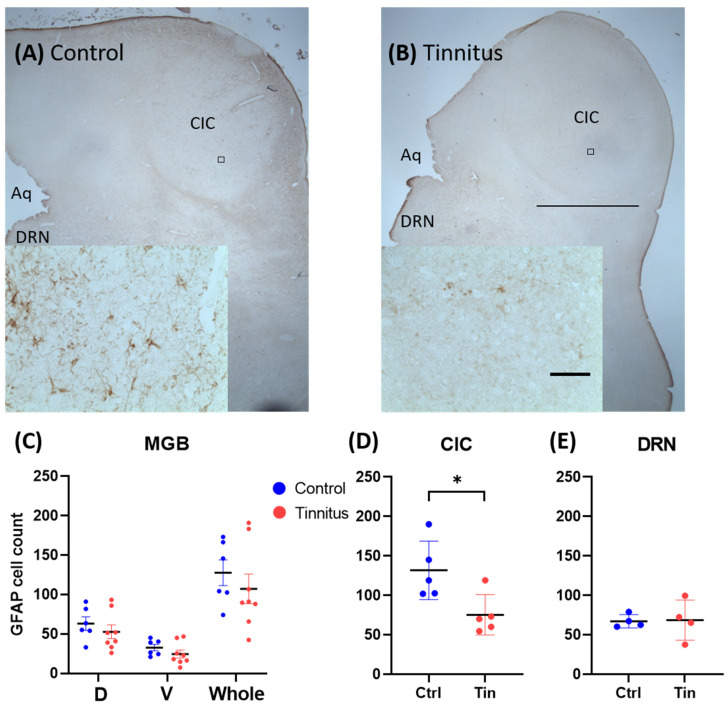
Glial fibrillary acidic protein (GFAP) expression in the MGB, CIC, and DRN. (**A**,**B**) Representative low-power and high-power photomicrographs inset in the left corner for control subject and tinnitus, respectively. (**C**–**E**) Cumulative data showing the means and SEMs of GFAP cell count in tinnitus and control groups. (**C**) Graph shows the number of GFAP-positive cells in (D) dorsal and (V) ventral and (whole) MGB. *p* value < 0.05 was defined as significant level and is indicated by a (*). Note the significant reduction in GFAP cell count in the CIC only. Scale bar = 2 mm; scale bar in the inset = 50 µm. Aq, cerebral aqueduct; CIC, central part of inferior colliculus; DRN, dorsal raphe nucleus; and MGB, medial geniculate body.

**Figure 4 brainsci-12-01024-f004:**
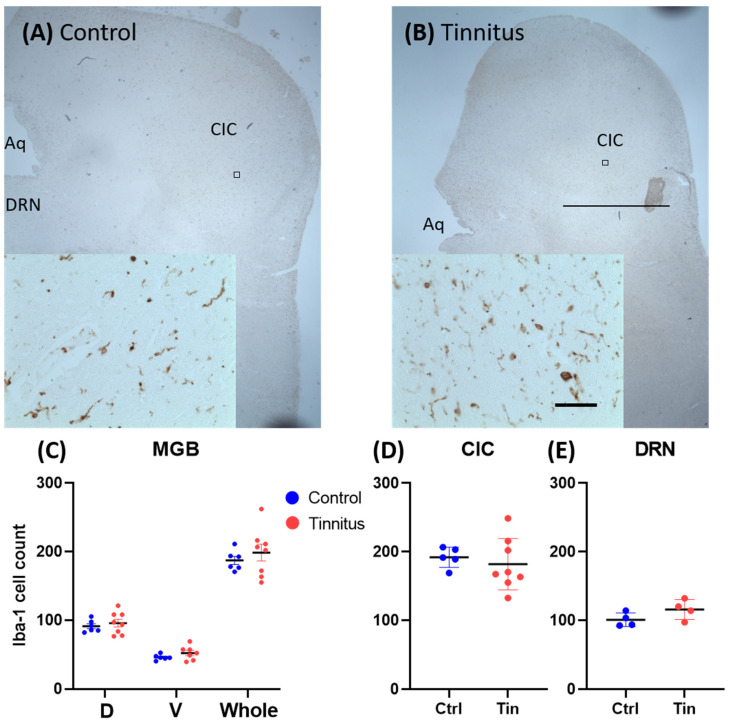
Iba-1 expression in the MGB, CIC, and DRN. (**A**,**B**) Photomicrographs of sections stained with Iba-1, taken from the adjacent sections shown in Figure 3. (**C**–**E**) Cumulative data showing the mean and SEMs of Iba-1 cell count. (**C**) Graph shows the Iba-1 cell count in the (D) dorsal, (V) ventral, and (whole) MGB in tinnitus and control subjects. *p* < 0.05 was defined as significant level. Note: no significant difference was found in any of the investigated regions when comparing the tinnitus group with controls. Scale bar = 2 mm; scale bar in the inset = 50 µm. Aq, cerebral aqueduct; CIC, central part of inferior colliculus; DRN, dorsal raphe nucleus; and MGB, medial geniculate body.

**Figure 5 brainsci-12-01024-f005:**
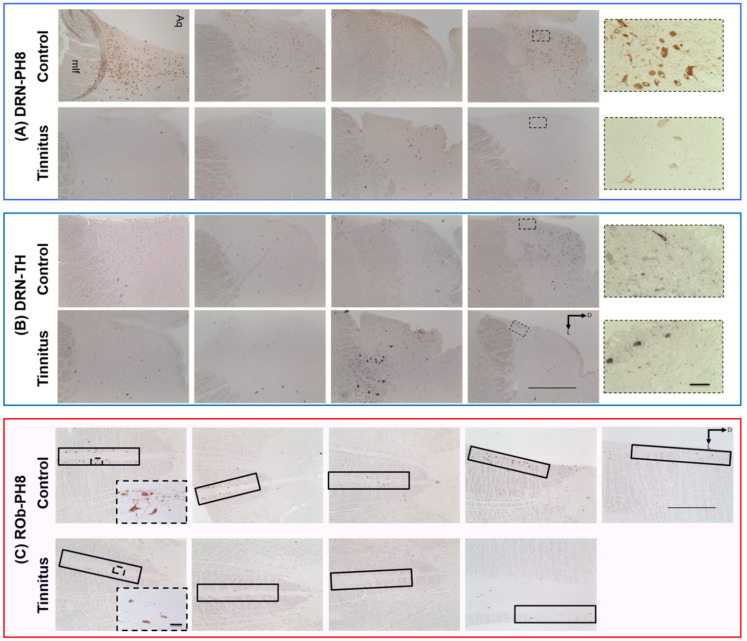
Representative low-power photomicrographs showing the brain sections containing the DRN (in blue rectangles) stained with PH8 (**A**) and adjacent sections stained with TH (**B**). A high-power photomicrograph of one subject (higher magnification of the dotted square) is shown in the right box in each line. Photomicrographs showing the ROb stained with PH8 (in red rectangle, (**C**)). Scale bar = 1 mm and for insets = 50 µm. Aq, cerebral aqueduct; D, dorsal; L, lateral; mlf, medial longitudinal fasciculus; Rob, obscurus raphe nuclei; TH, tyrosine hydroxylase; PH8, phenylalanine hydroxylase 8.

**Figure 6 brainsci-12-01024-f006:**
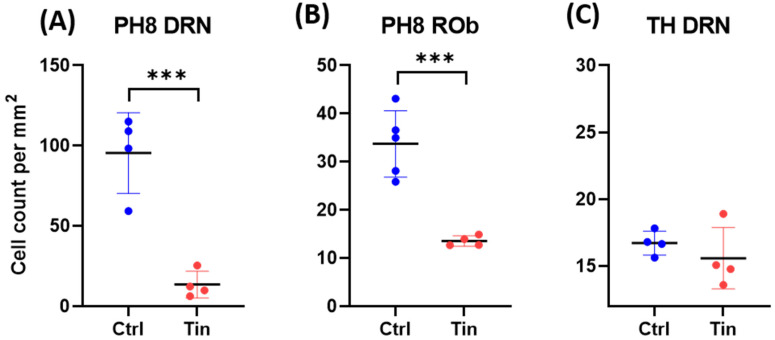
Cumulative data showing the means and standard error of means (SEMs) of PH8 (**A**,**B**) and TH (**C**) cell density in the DRN and ROb of tinnitus and control subjects. *p* value < 0.001 is indicated by a (***). Note the significant reduction in PH8 cell density in the DRN and Rob of tinnitus patients.

**Table 1 brainsci-12-01024-t001:** Specifications of control (C1–C8) and tinnitus (T1–T9) tissue samples.

Group	Case #	Duration	Disorder Characteristic	Sex	Age	pmd	Weight	MGB	TRN	LP	CIC	DRN	Rob
C1	S98/233			M	72	4	1385	√			√	√	√
C2	S11/091			M	76	7	1220	√	√				
C3	S14/063			F	93	8	1025	√	√			√	√
C4	S17/016			M	72	4	1385	√	√	√	√	√	√
C5	S17/093			M	82	6	1195	√	√		√	√	√
C6	PDC019			F	74	29	1071				√		√
C7	PDC033			F	77	43	1450	√	√	√			
C8	PDC086			M	89	34	1209	√	√	√	√		
							Total	7	6	3	5	4	5
T1	PDC028	-	B with HL	F	84	11	602 Hf	√		√			
T2	S00/157	5	B with HL	M	78	8	1222	√	√				√
T3	S00/255	8	B	F	94	7	937	√	√				√
T4	S01/119	7	U	M	75	6	1056	√	√		√	√	
T5	S10/033	4	U	M	70	6	1502	√	√	√	√	√	√
T6	C033	8	B with HL	M	75	18	1519	√	√	√	√	√	
T7	PDC015	long standing	B with HL	F	81	15	1352	√	√	√	√	√	
T8	PDC029	15	B with HL	M	82	48	656 Hf				√		√
T9	PDC032	30	B with HL	F	91	19	1068	√		√			
							Total	8	6	5	5	4	4

The duration (years) of the disorder was calculated from the onset of tinnitus in clinical records until age of death (where unknown, it is marked with a “-”). Table shows the laterality of tinnitus (B, bilateral; U, unilateral) and whether there was hearing loss (HL). M, male; F, female; pmd post-mortem delay; Hf, half fixed; MGB, medial geniculate body; TRN, thalamic reticular nucleus; LP, lateral nucleus of pulvinar; CIC, central part of inferior colliculus; DRN, dorsal raphe nucleus; and ROb, raphe obscurus nucleus.

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
