# Peer review of "Post-Mortem Analysis of Neuropathological Changes in Human Tinnitus"

_brainsci, 2022, doi:10.3390/brainsci12081024_

Round 1

Reviewer 1 Report

Almasabi et al. examined post-mortem brain tissues in tinnitus patients and compared differences with age-matched control groups. The study reports lower neuron density in MGB, CIC, lower astrocyte density in CIC, and lower serotonergic neuron density in DRN and ROb. These important results will no doubt contribute to the tinnitus literature. However, due to the limitation of post-mortem analysis and retrospective chart review, it remains unresolved whether the observed difference between tinnitus and control group is due to hearing loss (that may explain cell density in the auditory regions) or comorbidities of depression or anxiety (that may underlie differences in serotonergic system). While the authors acknowledge this, the statement does not occur until the last paragraph of the discussion. I urge the authors to disclose this information upfront in the introduction, so that readers can fully appreciate the complexity of tinnitus research. In addition, I only have a few questions and suggestions:

- The tinnitus samples are taken from patients who sought treatment primarily for tinnitus, and were prescribed hearing aids—is this correct? Or did they seek hearing aid for hearing loss with tinnitus as a comorbidity? In either case, what about the control samples? If the patients were seeking hearing aids, weren’t audiometric information like PTA recorded?

- It is worth noting in the method that the patients died of old age (correct?)

- Line 328: I suggest rephrasing to take other factors (the limitations, in particular) into account: it’s not so much difference between animal studies and human studies, it’s that the tinnitus is likely of different causes, i.e. noise-induced vs age-related.

- For the histology figure, please add “tinnitus” vs “control” labels. This will make the results much easier to read.

Author Response

Almasabi et al. examined post-mortem brain tissues in tinnitus patients and compared differences with age-matched control groups. The study reports lower neuron density in MGB, CIC, lower astrocyte density in CIC, and lower serotonergic neuron density in DRN and ROb. These important results will no doubt contribute to the tinnitus literature. However, due to the limitation of post-mortem analysis and retrospective chart review, it remains unresolved whether the observed difference between tinnitus and control group is due to hearing loss (that may explain cell density in the auditory regions) or comorbidities of depression or anxiety (that may underlie differences in serotonergic system). While the authors acknowledge this, the statement does not occur until the last paragraph of the discussion. I urge the authors to disclose this information upfront in the introduction, so that readers can fully appreciate the complexity of tinnitus research. In addition, I only have a few questions and suggestions:

Response: We like to thank the reviewer for his constructive criticism. We agree on the importance of hearing loss as a potential confounder. As suggested we highlighted this further in the introduction as well as in the methods section. We also would like to point to the high co-occurrence of these comorbidity in persons with tinnitus. The association of hearing loss with tinnitus reach up to 96.9% (Manche, et al. 2016).

- The tinnitus samples are taken from patients who sought treatment primarily for tinnitus, and were prescribed hearing aids—is this correct? Or did they seek hearing aid for hearing loss with tinnitus as a comorbidity? In either case, what about the control samples? If the patients were seeking hearing aids, weren’t audiometric information like PTA recorded?

Records were checked from persons that donated their brain to the Netherlands Brain Bank and the Parkinson's United Kingdom Brain Bank. It was not clarified how this diagnosis was made. It is thus also unclear if and what kind of treatment was provided. Audiometric information was not available.  We added this to the methods section.

- It is worth noting in the method that the patients died of old age (correct?)

Persons generally died of old age. In most cases a more specific cause of death is mentioned such as heart attack, cancers or pneumonia. We included a sentence about this in the methods section. For the Netherlands Brain Bank no records on the cause of death are made available for the control group.

- Line 328: I suggest rephrasing to take other factors (the limitations, in particular) into account: it’s not so much difference between animal studies and human studies, it’s that the tinnitus is likely of different causes, i.e. noise-induced vs age-related.

Implemented.

- For the histology figure, please add “tinnitus” vs “control” labels. This will make the results much easier to read.

We added the labels as suggested.

Reviewer 2 Report

Line 42-43 "Tinnitus is a sensory disorder in which a phantom sound is perceived by affected 42 individuals. It is estimated that approximately 15% of the general population experience 43 tinnitus": authors did not report any reference for such data

Line 49-51: authors report a reference by Eggermont et al 2015 referring to EEG abnormalities as a consequence of tinnitus, however it is interesting to note that tinnitus related abnormalities has been eveidenced also at "lower" stages of the auditory system, that is brainstem (Cartocci, G., Attanasio, G., Fattapposta, F., Locuratolo, N., Mannarelli, D., & Filipo, R. (2012). An electrophysiological approach to tinnitus interpretation. The international tinnitus journal17(2), 152-157; Edvall, N. K., Mehraei, G., Claeson, M., Lazar, A., Bulla, J., Leineweber, C., ... & Cederroth, C. R. (2022). Alterations in auditory brain stem response distinguish occasional and constant tinnitus. The Journal of clinical investigation132(5); Milloy, V., Fournier, P., Benoit, D., Noreña, A., & Koravand, A. (2017). Auditory brainstem responses in tinnitus: a review of who, how, and what?. Frontiers in aging neuroscience9, 237.)

Author Response

Line 42-43 "Tinnitus is a sensory disorder in which a phantom sound is perceived by affected 42 individuals. It is estimated that approximately 15% of the general population experience 43 tinnitus": authors did not report any reference for such data

Thank you for your keen eye. We added the reference and provided a bit more insight on what these 15% means.

Line 49-51: authors report a reference by Eggermont et al 2015 referring to EEG abnormalities as a consequence of tinnitus, however it is interesting to note that tinnitus related abnormalities has been eveidenced also at "lower" stages of the auditory system, that is brainstem (Cartocci, G., Attanasio, G., Fattapposta, F., Locuratolo, N., Mannarelli, D., & Filipo, R. (2012). An electrophysiological approach to tinnitus interpretation. The international tinnitus journal17(2), 152-157; Edvall, N. K., Mehraei, G., Claeson, M., Lazar, A., Bulla, J., Leineweber, C., ... & Cederroth, C. R. (2022). Alterations in auditory brain stem response distinguish occasional and constant tinnitus. The Journal of clinical investigation132(5); Milloy, V., Fournier, P., Benoit, D., Noreña, A., & Koravand, A. (2017). Auditory brainstem responses in tinnitus: a review of who, how, and what?. Frontiers in aging neuroscience9, 237.)

We like to thank the reviewer for highlighting these interesting papers and did read them with interest. We, however, decided not to include these findings into the introduction of this manuscript as this would draw way the attention to the scope of our paper.

Reviewer 3 Report

Dear authors,

My most sincere congratulations for this work. Post-mortem neuropathological studies are getting infrequent (due to different reasons), too many studies in animal models are leaving behind post-mortem studies. In this work, regions such as the medial geniculate body, the thalamic reticular nucleus, the inferior colliculus, the dorsal and obscurus raphe nuclei have been examined in search of neuropathological changes underlying tinnitus.

Here below some comments:

-It would be of high interest to know which fixative has been used, how many time the samples have been left in the fixative and which preservation protocol (how often the fixative has been changed) and sampling protocol have been followed. Have brains been immersion fixed? 

I understand that the samples have been provided by two different brain banks, but I think that this is a key point to know for researcher in the field.

-On the other side, in table 1 you specify the post-mortem delay for each case. How have the cadavers been preserved before autopsy?

-Figure 1: FWM, not explained

-Figure 5 organization is quite confused and difficult to follow.

For future studies, you can consider to work with a higher thickness of the tissues (you can check the following paper for big brains: https://www.ncbi.nlm.nih.gov/pmc/articles/PMC8879147/), cryoprotecting the samples, cutting them in a cryostat or sliding freezing microtome, and thus performing stereological cell counting (and avoiding the antigen retrieval step).

Kind regards and congratulations.

Author Response

Dear authors,

My most sincere congratulations for this work. Post-mortem neuropathological studies are getting infrequent (due to different reasons), too many studies in animal models are leaving behind post-mortem studies. In this work, regions such as the medial geniculate body, the thalamic reticular nucleus, the inferior colliculus, the dorsal and obscurus raphe nuclei have been examined in search of neuropathological changes underlying tinnitus.

Thank you for your kind words and constructive feedback.

Here below some comments:

-It would be of high interest to know which fixative has been used, how many time the samples have been left in the fixative and which preservation protocol (how often the fixative has been changed) and sampling protocol have been followed. Have brains been immersion fixed? 

I understand that the samples have been provided by two different brain banks, but I think that this is a key point to know for researcher in the field.

The included samples were retrieved from the UK and Netherlands Brain banks. Post-mortem processing, including fixation and preservation, of the brains was conducted by them. We received the tissue as blocks and performed cutting, staining, and quantifications as explained in the methods section. We consider that all brain tissue was processed by the same standard protocol for each brain bank in those Formalin-fixed tissue blocks. Effect of tissue processing on our results can not completely be ruled out. To asses this, we conducted some additional test/ steps to minimize the impact of tissue handling:

  • First, we included a comparable number of samples from tinnitus vs controls in each brain bank. As each bank may differ in their specification methods of fixation, including similar number of tinnitus and control samples per region reduced this possibility.
  • Next, we included control regions for all stainings whenever possible. For Nissl, we added the lateral nucleus of pulvinar (LP) region . The LP is a visual thalamic region, lies next to the MGB. The LP did not show any differences in cell count. We therefore concluded that the cell reduction was limited to auditory regions specifically and not a results of tissue handling. Moreover, if results could be explained by tissue handling only, we would expect that a reduction would be observed in all investigated regions including the  TRN and DRN, which showed no significant difference.
  • In all related stainings (GFAP/Iba-1, and PH8/TH) we used consecutive sections. This allows better evaluations of these changes since those are linked stainings. Also, for glial cells evaluation, they were also a consecutive section to Nissl staining ( e.g. Nissl s1, GFAP s2, Iba-1 s3). Thus, mostly those section represent overlapped or very close cell population due to limited thickness (7µm). This reduced the possibility of fixation issue as contributed factor as some of this changes were not significant and in some cases increased number -but not significant- were observed, see Iba-1 in MGB and TRN.

We further clarified this in the methods section.

-On the other side, in table 1 you specify the post-mortem delay for each case. How have the cadavers been preserved before autopsy?

Unfortunately no information is available about the time between death and preservation of the tissue.

-Figure 1: FWM, not explained

Thank you for your keen eye. We added this to the figure caption

-Figure 5 organization is quite confused and difficult to follow.

Thanks for pointing our attention to this issue. We therefore, made some changes on the orientation. Specially, we align each staining per region together in the same box, so a compassion between groups become easier.  

For future studies, you can consider to work with a higher thickness of the tissues (you can check the following paper for big brains: https://www.ncbi.nlm.nih.gov/pmc/articles/PMC8879147/), cryoprotecting the samples, cutting them in a cryostat or sliding freezing microtome, and thus performing stereological cell counting (and avoiding the antigen retrieval step).

Thank you for pointing us to this manuscript. We will definitely consider this in future studies.

Kind regards and congratulations.